# Intra-Articular Injection of Botulinum Toxin for the Treatment of Knee Osteoarthritis: A Systematic Review of Randomized Controlled Trials

**DOI:** 10.3390/ijms24021486

**Published:** 2023-01-12

**Authors:** Cristiano Sconza, Giulia Leonardi, Carla Carfì, Elizaveta Kon, Stefano Respizzi, Dalila Scaturro, Giulia Letizia Mauro, Giuseppe Massazza, Berardo Di Matteo

**Affiliations:** 1Department of Biomedical Sciences, Humanitas University, Pieve Emanuele, 20072 Milan, Italy; 2IRCCS Humanitas Research Hospital, Rozzano, 20089 Milan, Italy; 3U.O.C. of Physical and Rehabilitation Medicine and Sports Medicine, Policlinico Universitario “G. Martino”, 98124 Messina, Italy; 4Department of Surgical, Oncological and Stomatological Disciplines, University of Palermo, Via del Vespro, 129, 90127 Palermo, Italy; 5Division of Physical Medicine and Rehabilitation, Department of Surgical Sciences, University of Turin, 10124 Turin, Italy

**Keywords:** botulinum, musculoskeletal, intra-articular, arthritis, tendinopathy, systematic review

## Abstract

The purpose of the present paper was to review the available evidence on intra-articular botulinum toxin (BTX) injection in the treatment of knee osteoarthritis and to compare it to other conservative treatment options. A systematic review of the literature was performed on the PubMed, Scopus, Cochrane Library, Web of Science, Pedro and Research Gate databases with the following inclusion criteria: (1) randomized controlled trials (RCTs), (2) written in the English language, and (3) published on indexed journals in the last 20 years (2001–2021) dealing with the use of BTX intra-articular injection for the treatment of knee OA. The risk of bias was assessed using the Cochrane Risk of Bias tool for RCTs. Nine studies involving 811 patients in total were included. Patients in the control groups received different treatments: conventional physiotherapy, hyaluronic acid injection or prolotherapy or a combination thereof in 5 studies, steroid infiltrative therapy (triamcinolone) in 1 study, placebo in 2, and local anesthetic treatment in 1 study. Looking at the quality of the available literature, two of the included studies reached “Good quality” standard, three were ranked as “Fair”, and the rest were considered “Poor”. No major complications or serious adverse events were reported following intra-articular BTX, which provided encouraging pain relief, improved motor function, and quality of life. Based on the available data, no clear indication emerged from the comparison of BTX with other established treatments for knee OA. The analysis of the available RCTs on BTX intra-articular injection for the treatment of knee OA revealed modest methodological quality. However, based on the data retrieved, botulinum toxin has been proven to provide good short-term outcomes, especially in patients with pain sensitization, by modulating neurotransmitter release, peripheral nociceptive transduction, and acting on the control of chronic pain from central sensitization.

## 1. Introduction

Osteoarthritis (OA) is a widespread musculoskeletal disease and a leading cause of chronic disability [1]. Conservative estimates state that up to 240 million people worldwide suffer from it [2]. Knee osteoarthritis (KOA) is associated with chronic inflammation that causes persistent oxidative damage, which subsequently leads to joint degeneration [3,4]. The resulting functional limitation and reduced physical activity increase the risk of psychological distress and reduce quality of life [5]. Therapeutic options have increased and improved in recent years; nevertheless, there is not yet an ideal treatment for KOA [6]. The current management of KOA follows a stepwise approach [7]. Nonpharmacological approaches are recommended as first-line treatment, including exercise and weight loss [8,9]. When conservative treatment is not enough, oral analgesics, such as non-steroidal anti-inflammatory drugs (NSAIDs), can be administered, providing temporary pain relief [7]. Unfortunately, there are some safety concerns that limit their suitability for long-term administration [10]. Thus, there is a need for effective and well-tolerated treatment options for long-term pain management in patients with KOA, especially for those unsuitable for surgical management [11]. Several treatment options for KOA have been investigated, and minimally invasive strategies, such as intra-articular injections, have been shown to be well tolerated and able to provide good clinical results [12]. Beyond traditional options such as corticosteroids and hyaluronic acid (HA), whose role has been recognized over time by medical societies, novel biologic agents have also been introduced into clinical practice, such as platelet-rich plasma (PRP) and mesenchymal stem cells, with the aim of modulating the joint environment through the action of bioactive molecules and growth factors able to counteract inflammatory stimuli and promoting tissue repair and regeneration. Although favorable outcomes are described in literature, these biologic treatments cannot be considered first-line treatments, and there is still a lack of clear guidelines on their application [13,14,15].

Looking at other approaches, results from recent studies suggested that botulinum toxin may also have a role in nociceptive pain [11]. Botulinum toxin (BTX) is a multi-molecular complex produced by various strains of the anaerobic bacterium Clostridium Botulinum [16,17]. The neurotoxins are associated with complexing proteins that protect them from degradation [18,19]. The intra-articular administration of Botulinum neurotoxin type A may inhibit the release of inflammatory mediators and neuropeptides from the nociceptors, thereby reducing the pain and the neurogenic inflammation that occurs in OA [20,21]. BTX might also have an anti-nociceptive effect by down-regulating the voltage-gated Na+-channel expression [22], as proved in the rat trigeminal neuralgia [23], or by reducing the peripheral release of neurotransmitters (e.g., substance P and CGRP) and pro-inflammatory cytokine IL-1β [24]. Moreover, it inhibits the fusion of intracellular vesicles with nerve membranes, thus further impairing the release of neurogenic inflammatory mediators [11]. Researchers have reported that a single intra-articular BTX injection improves symptoms in some KOA patients with chronic and refractory pain but not in others, suggesting that different sub-groups may exist [25]. Based on this rationale, the off-label use of botulinum toxin could provide a new approach for the treatment of KOA in orthopedic settings [26].

The purpose of the present paper is to systematically review the available high-quality evidence on the application of BTX injections for the treatment of KOA in order to understand its real therapeutic potential and to compare it with other conservative treatment options.

## 2. Methods

A systematic review of the literature was performed on the use of intra-articular botulinum toxin injections as knee osteoarthritis treatment. We conducted the search for English articles published up to the end of July 2022 according to the Preferred Reporting Items for Systematic Reviews and Meta-Analyses (PRISMA) principles. The electronic databases PubMed, Scopus, Cochrane Library, Web of Science, Pedro, and Research Gate were investigated, using the following key words, which were combined to achieve maximum search strategy sensitivity: (“botulinum”) AND (“musculoskeletal” OR “intra-articular” OR “arthritis” OR “tendinopathy”). We integrated the database search with the screening of the reference lists and the monitoring of citations included in the studies to identify any additional studies. An additional search was performed on nonconventional databases (i.e., the so-called “gray literature”). Two independent observers (CS and GL) conducted the screening process and the analysis separately. 

First, articles were screened by title and abstract, using the following inclusion criteria for selection: (1) randomized controlled trials (RCTs); (2) written in English language; (3) published on indexed journals from 2001 to 2022; and (4) dealing with the use of intra-articular botulinum neurotoxin for the treatment of KOA. The exclusion criteria were (1) non-randomized trials; (2) reviews; (3) papers written in other languages than English; and (4) data not dealing with the treatment of knee osteoarthritis. Second, the full texts of the selected articles were screened with further exclusions according to the previously described criteria. A PRISMA flowchart of the selection and screening method is provided in Figure 1.

We extracted and collected the relevant data in a single database with the agreement of the two observers: (1) treatment groups, (2) sample size and patients’ features, (3) BTX preparation method, (4) therapeutic protocols, (5) outcome measures, (6) timepoints of follow-up evaluations, and (7) a summary of clinical results. Any divergence was discussed with the senior investigator (BDM), who made the final judgement.

The risk of bias was assessed using the Cochrane Risk of Bias tool for Randomized Controlled Trials, which evaluates seven different types of bias. Each of them, based on specific criteria, was classified as “Low risk”, “High risk”, or “Unclear risk”. Subsequently, the results of this assessment were converted to AHRQ (Agency for Healthcare Research and Quality) Standards, which ultimately rank the RCTs in “Good quality”, “Fair quality”, and “Poor quality”.

## 3. Results and Discussion

### 3.1. Results

A total of 9 studies, published from 2009 to July 2022, dealing with intra-articular BTX injection for KOA were included in this review. A detailed description of each study has been provided in Table 1.

#### 3.1.1. Study Design and Quality

According to the inclusion criteria, all studies were randomized controlled trials. The study designs were highly variable, since patients in the control groups received different injections or treatments: corticosteroids (CS) in 2 studies [20,27]; simulated injections in 2 studies [11,25]; simulated injections + anesthetics drugs in 1 study [26]; physical therapy in 1 study [28]; education for arthritis care in 1 study [29]; Hyaluronic acid + Dextrose prolotherapy + physical therapy in 1 study [5]; and Hyaluronic acid + physical therapy in another study [30].

Looking at the quality of the available literature by the AHRQ standard, we found that two studies reached a “good quality” standard, whereas 3 were ranked as “fair quality”, and the rest were considered “poor quality”. The results of the analysis performed with the Cochrane Risk of Bias tool for RCT are detailed in Table 2. The random sequence generation process was specified in 8 papers [5,11,20,25,27,28,29,30]. The method of allocation concealment was described in only 5 of the studies included [5,11,20,25,29]. Regarding sample size calculation, the power analysis methods were not fully clarified in three trials only [20,26,30]. Four trials were double-blinded [11,20,25,27], three were single-blinded [28,29,30], and the others were unblinded. Moreover, the risk of attrition bias was low for the majority of the studies; in all the included studies, it was clearly specified the number of patients screened; how many were excluded from randomization; and why, how many were lost to follow-up, and for what specific reason. Flow diagrams depicting the patients’ selection process were reported in all included studies except for one [26]. Finally, we found that three protocol trials were not registered in a public registry [26,27,29], which should be mandatory according to the Consolidated Standards of Reporting Trials (CONSORT) 2010 guidelines.

**Table 1 ijms-24-01486-t001:** The use of botulinum toxin in the treatment of KOA: data extracted from the 9 RCTs included in the review.

Publication	Study Design	Pathology	Score	Patients Features	BoNT-A Preparation Method	Therapeutic Protocol and F-up	Results	Overall Performance of BTX
Rezasoltani et al., 2021 [28]	Single-blind RCT (BTX injection vs. physical therapy [PT])	knee osteoarthritis	VAS, KOOS	50 (25 vs. 25) Age: 77.7 ± 7.3 y 63.0 ± 8.0 y Sex: F 73%: F 80%	V & Conc: 100 IU of BTX (250 units from disport brand) in 5 mL of SS	F-up at 1, 3, and 6 mo	At 1 mo F-up, VAS score and all KOOS subscales were improved in the BTX group in comparison to the PT group. The use of BTX can reduce pain and improve the function and quality of life in patients with KOA.	BTX+
Rezasoltani et al., 2020 [5]	RCT (Physical therapy vs. BTX injection vs. Hyaluronic acid vs. Dextrose prolotherapy)	knee osteoarthritis	VAS, Persian version of KOOS	120 (30 vs. 30 vs. 30 vs. 30) Age: 70 (±6.3) y 67.7 (±7.3) y 66.1 (±9.1) y 64.8 (±5.8) y Sex: 12 M:18 F 8 M:22 F 14 M:16 F 11 M:19 F	V & Conc: 250 units of Dysport, equivalent to 100 units of BoNT/A (Dysport, Abobotulinumtoxin A), diluted with 5 mL of SS	F-up at baseline, and in 1 wk, 4 wks, and 3 mo	An IA injection of BTX or dextrose prolotherapy is effective first-line treatments. In the next place stands physical therapy particularly if the patient is not willing to continue regular exercise programs. The study was not very supportive of IA injection of hyaluronic acid as an effective treatment of KOA	BTX+
Mendes et al., 2019 [20]	Double-blind RCT (BTX injection vs. TH injection vs. placebo)	knee osteoarthritis	VASm, VASr, WOMAC, 6-min walk test, TUG, SF-36, ROM of knee and US measurement of synovial hypertrophy.	105 (35 vs. 35 vs. 35) Age: (64.2 ± 6.9 y) Sex: 9 M:96 F	V & Conc: 100 IU of BTX in 2 mL of SS 0.9%	F-up at baseline and at 4, 8, and 12 wks	IA injection with TH in primary KOA had a higher effectiveness than that with BTX or SS in the short-term assessment (4 wks) for VASm, WOMAC, and US measurement of synovial hypertrophy	BTX-
McAlindon et al., 2018 [11]	Double-blind RCT (BTX injection vs. placebo)	knee osteoarthritis	NRS, WOMAC pain and physical function scores, PGIC	176 (44 vs. 43 vs. 89) Age: 60.7± 8.3 y 60.2 ± 8.4 y 61.1± 7.8 y Sex: 30 (68.2%) F 26 (60.5 %) F 51 (57.3%) F	V & Conc: 400 or 200 IU of BTX in a total volume of 2 mL	F-up at at wks 1 and 4 and every 4 wks thereafter to wk 24 were undertaken	There were no significant differences between IA BTX and placebo in reducing WOMAC pain and physical function scores at wk 8 compared with baseline, in patients with KOA and nociceptive pain	BTX=
Bao et al., 2018 [30]	Single-blind RCT (BTX injection + therapeutic exercises vs. Hyaluronate injection + therapeutic exercises vs. placebo + therapeutic exercises)	knee osteoarthritis	VAS, WOMAC and SF-36	60 (20 vs. 20 vs. 20) Age: 66.4 ± 3.49 y 66.0 ± 2.09 y 65.3 ± 3.52 y Sex: 10 M:10 F 13 M:7 F 9 M:11 F	V & Conc: 100 IU BTX (Botox; Allergan Inc., Irvine, KY, USA) diluted with 2.5 mL preservative- free 0.9% SS	F-up at baseline, and at the end of the 4th and 8th wks	At the end of the 4th and 8th wks, WOMAC and VAS scores were higher in the CG. Therapeutic exercise plus BTX or hyaluronate injection can significantly reduce pain and improve knee functioning in patients with KOA. BTX plus therapeutic exercise appears to be more effective.	BTX+
Hsieh et al., 2016 [29]	Single-blind RCT (BTX injection vs. education only for arthritis care)	knee osteoarthritis	VAS, LEQUESNE and WOMAC indexes	46 (21 vs. 20) Age: 67.82 ± 9.06 y 68.06 ± 4.53 y Sex: 32: F (52.5%) 30: F (50.0%)	V & Conc: 100 IU of BTX (Botox, Allergan Inc., Parsippany, NJ, USA) diluted with 2 mL of preservative- free 0.9% SS	F-up at 1 wk and 6 mo	The pain VAS score in the BTX group significantly decreased at 1 wk and at 6 mo post treatment but not in the CG. Significant differences for the between-group comparison were observed in WOMAC and Lequesne indexes at 6 mo f-up. The IA injection of BTX provided pain relief and improved functional abilities in patients with KOA in both the short- and long-term f-up	BTX+
Arendt-Nielsen et al., 2016 [25]	Double-blind RCT (BTX injection vs. placebo)	knee osteoarthritis	NRS, WOMAC, ADP, GIC	121 (61 vs. 60) Age: 62.5 ± 8.6 y 62.1 ± 8.6 y Sex: 23:M 15:F 23:M 14:F	V & Conc: 200 IU of BTX containing 2 mL of 0.9% SS	F-up at baseline and weeks 4, 8, and 12	The nociceptive group showed significant improvement after IA BTX at wk 8 for all WOMAC outcomes, ADP at wks 9 and 10, and patient GIC at wk 12. IA BTX given to patients with nociceptive KOA reduced pain sensitization together with improvement in pain and function	BTX=
Boon et al., 2010 [27]	Double-blind RCT (Low-dose of BTX injection vs. High-dose of BTX injection vs. CS injection)	knee osteoarthritis	VAS, WOMAC, SF-36, PGA, 40-m timed walk	60 (20 vs. 20 vs. 20) Age: 64.1 ± 13.4 y 61.2 ± 9.4 y 60.8 ± 10.1 y Sex: 9/11: M (45%) 9/11: M (45%) 7/13: M (35%)	V & Conc: 100 IU of BTX, 200 IU of BTX	F-up at baseline, 4, 8, 12, and 26 wks	At 8 wks, VAS score decreased within each group but only reached statistical significance in the low-dose BTX group. All groups showed statistically significant improvements in all WOMAC scores at 8 wks. Possible role for BTX as a treatment option for moderate pain and functional impairment secondary to KOA.	Low dose of BTX+
Mahowald et al., 2009 [26]	RCT (BTX+ Lidocaine injection vs. saline placebo + Lidocaine)	shoulder and knee osteoarthritis	VAS, WOMAC, SF-MPQ, SF-36	78 (36 vs. 42) (Shoulder study vs. Knee study) Age: NA Sex: NA	V & Conc: 25–100 IU of BTX with 2 cc of 2% Bupivacaine	F-up at baseline, at 1 and 3 mo	In the shoulder study, IA-BTX produced a significant decrease in shoulder pain severity at 1 mo that was also significantly better than the non-significant change after IA-Saline placebo. In the knee study IA-BTX produced a significant 48% decrease in SF-MPQ at 1 mo that was still significant at 3 mo after injection. There was a strong placebo response in 1/3 of those but the decrease in pain severity was not significant.	BTX=

ADP, average daily pain; BTX, Botulinum toxin; CS, corticosteroids; PA, patient; CG, control group; Conc.,concentration; F, female; FU, follow-up; GIC, global impression of change; IA, intrarticular; KOA, knee osteoarthritis; KOOS, The Knee injury and Osteoarthritis Outcome Score; LD, large dose group; M, male; MO, month; NA, not available; NRS, numerical rating scale; OA, osteoarthritis; PS, pain scores; PGA, patient global assessment; PGIC, patient global impression of change; RCT, randomized controlled trial; ROM, range of motion; SD, small-dose group; SF-36, the 36-Item Short Form Health Survey; SF-MPQ, Short-form McGill Pain Questionnaire; SG, study group; SS, saline solution; TH, triamcinolone hexacetonide; TG, Treated Group; PG, Placebo Group; TUG, A Timed “Up-and-Go” test; US, ultrasound; V, volume; VAS, Visual analog scale; VASm, VAS for pain during movement; VASr, VAS for pain at rest; wk, week; WOMAC, Western Ontario and McMaster Universities Osteoarthritis Index.

#### 3.1.2. Patients and Evaluation Methods

Nine studies involving a total of 811 patients with KOA were included. The mean age was 65 years. In most papers, the diagnosis of KOA was assessed using the American College of Rheumatology criteria, documented by patients’ medical record, interviews, physical examination, and confirmed with a radiological examination (Kellgren-Lawrence score from grade I to IV).

Baseline and follow-up assessments were based on clinical and radiological scores. The most used clinical scores were the Visual Analogue Scale (VAS), the Knee Injury and Osteoarthritis Outcome Score (KOOS), and the Western Ontario and McMaster Universities Osteoarthritis Index (WOMAC). Two trials reported also X-ray and MRI outcomes [29,30]; one trial included the assessment of synovial hypertrophy, measured by an ultrasonography of the infrapatellar recess and expressed using a quantitative grayscale [20].

#### 3.1.3. Treatment

The volume and concentration of BTX were quite uniform, and in the majority studies, the intra-articular dosage was 100 UI diluted with 2 mL of saline solution (Table 1). Only two papers reported a different volume dosage of BTX (200 UI) [11,25]. In all the studies, the BTX was used as a single intra-articular injection. The treatment protocols were very similar in terms of the number of injections and frequency (Table 1).

#### 3.1.4. Complications

BTX could denervate cholinergic sympathetic and parasympathetic neurons and affect autonomic functions including salivation, sweating, heart rate, and vasodilatation [31]. No major complications were reported in any of the studies considered. Rezasoltani et al. described two patients in the BTX group experiencing severe knee pain after injection [28]. MacAlindon et al. reported treatment-emergent adverse events (TEAEs) in the BTX group, such as arthralgia and swelling [11].

#### 3.1.5. Reported Clinical Outcome

Rezasoltani et al. demonstrated that BTX could reduce pain and improve the function and quality of life of patients with KOA; at 1 month F-up, VAS-score and all KOOS subscales were improved in the BTX group compared to the PT group [28]. The same results were found in another Rezasoltani et al. study, demonstrating that BTX injection and dextrose prolotherapy both worked well as first-line treatment for KOA while HA was the least effective therapy [5].

Hsieh et al. found a higher success rate in the BTX group compared to education for OA care, with significant improvement in pain and functional abilities at short (1 week) and long-term (6 months) follow-up [29]. Conversely, Mendes et al. compared the effects of triamcinolone hexacetonide injection (TH) vs. BTX and vs. saline solution injection (SS), demonstrating that TH had a better effect in short-term (4 weeks) improvement in the VAS, WOMAC, and US measurement of synovial hypertrophy [20]. MacAlindon found no statistically significant differences between the effects of intra-articular BTX and the placebo in reducing pain at 8 weeks in patients with KOA [11]. Arendt-Nielsen’s study reported no significant difference in clinical outcome between BTX and the placebo, although intra-articular BTX seemed to have an anti-hyperalgesic effect after 4 weeks and reduced pain sensitization in patients with nociceptive KOA [25]. Bao’s study demonstrated that intra-articular BTX, in addition to therapeutic exercise, is more effective than HA injection, reducing pain and improving knee function at 4 and 8 weeks [30]. In the pilot study of Boon et al., subjects were randomized to receive a single injection of corticosteroid, low-dose Botulinum toxin type A (100 units), or high-dose Botulinum toxin type A (200 units). The primary end point was a pain VAS score at 8 weeks, which decreased within each group but only reached statistical significance in the low-dose BTX [27]. Finally, Mahowald et al. documented that intra-articular BTX produced a significant decrease (48%) in SF-MPQ up to 3 months after injection; there was also a response to the placebo in 1/3 of patients, but it was not statistically significant as in the BTX group [26].

### 3.2. Discussion

The main findings of the present systematic review were (1) overall “fair” quality of studies comparing the use of botulinum toxin with other treatments and (2) the lack of univocal results for the intra-articular use of BTX in the treatment of KOA.

BTX was tested against the placebo, exercise therapy, pharmacological agents, and other common injections, such as corticosteroids and HA; unfortunately, the low number of studies found, with different clinical scores adopted, did not allow the authors to perform a meta-analysis of the results. Most of the RCTs analyzed in this review are characterized by a small simple size and weak power analysis, in some cases lacking a clear indication of numerical data used to calculate the sample size, which is therefore at high risk of being underpowered with obvious consequences for the significance of the results. In addition, there is a general partial adherence to CONSORT guidelines for reporting methods and results in RCTs, thus generating a series of consecutive biases responsible for the modest rating of the studies according to the AHQR standard: only two of them, in fact, could be rated as “Good Quality” RCTs. Despite the aforementioned methodological limitations, some clinical consideration can be drawn from the analysis of the literature, which underlines doubtful results of BTX therapy in reducing pain and improving the functional status in some specific patients affected by knee OA. Overall, five out of nine studies showed a more effective outcome for BTX treatment than the control group, whereas three studies showed no significant differences to the placebo, a finding that certainly needs to be investigated further. Only one study showed greater efficacy of the control treatment (steroid injection) compared to BTX.

The treatment protocols employed in the various trials were overall quite similar; most Authors opted for single administration of intramuscular BTX with a dosage of 100 UI [5,20,26,27,28,29,30], whereas some tried higher concentrations, for example 200 UI [11,25,27] or 400 UI [11]. Boon et al. compared the efficacy and safety of IA injections of low-dose Botulinum toxin type A (100 units) to high-dose Botulinum toxin type A (200 units) in patients with symptomatic knee OA, showing that changes in the pain score were significant only in the low-dose Botulinum toxin type A group [27]. Toxicological studies have reported that the human median lethal dose is approximately 2800 units, equivalent to 28 individual vials of Botox Purified Neurotoxin Complex (100 Units) for a 70-kg adult [32]. Therefore, considering intra-articular injections, 100 units seems to be safe and effective [29].

An interesting issue involves the correct indication for the use of this treatment: which patients can benefit most from BTX? Considering the study selection criteria, the majority of authors have treated patients with moderate-to-severe knee OA. Only 2 studies also considered patients with Kellgren-Lawrence (K-L) grade 1 [24,26], and both showed no significant differences in clinical efficacy between BTX and the placebo. In some cases [31], patients with rheumatologic diseases, K-L grade 4 [30] osteoarthritis, and the presence of marked joint effusion at the time of injection have also been included. The different inclusion criteria complicates the analysis on the specific indications for treatment. As far as it can be inferred, BTX use seems more indicated in patients with moderate grade knee OA.

The BTX effects in pain relief and functional improvement was evaluated mostly at short term with a consequent lack of data on the real duration of the beneficial effects. Only a few papers evaluated patients with a follow-up after more than 3 months, still demonstrating the persistence of the positive results of this therapy. Mahowald et al. reported that the pain decrease could last up to 12 months [26]. Singh et al. noted that the effect of repeated BTX injections could last for up to 17 months [32]. Heish et al. concluded that the duration of the antinociceptive effects of BTX remains unknown, perhaps due to the fact that there are more analgesic mechanisms involved, some of them independent from the mere neuromuscular junction blocking action in cholinergic nerves [29].

An advantage of using BTX injections is the possibility of a single administration. Conversely, other products such as hyaluronic acid, ozone, and PRP require multiple injections, with an obvious increase in the procedure-related risks. Looking at the comparison with corticosteroids, BTX does not cause metabolic impairment, does not provoke cartilage degeneration or bone ischemia, and can also be used in metal-replaced joints. On the other side, the cost of botulinum toxin type A is much higher [20]. Another positive aspect is that no major complications or serious adverse events connected to the treatment were reported in any of the studies. The most common complication of BTX is muscle weakness. When injected intra-articularly, BTX directly acts on the peripheral pain receptors, and it does not reach blood circulation or get absorbed by muscles. Thus, an IA injection appears to be safer than an intra-muscular injection [29]. Other potential complications include arrhythmia, dysphagia, anaphylactic shock, and skin rash. Such adverse events were not observed in the patients considered in the present review. The most frequent adverse effect was pain in the days immediately following the injection with subsequent spontaneous resolution [11,27].

Getting back to the previous question concerning patients who might better respond to intra-articular BTX therapy, an interesting insight came from the study by Arendt-Nielsen et al.; they demonstrated the effectiveness of the treatment specifically in patients with prevalent nociceptive pain, who experienced a reduction in symptoms together with improvement in joint function. Neurogenic inflammatory mediators are abundant in the sensory nerve endings of osteoarthritic knees, and they may sensitize peripheral nociceptors, thus generating more nociceptive firing and facilitating widespread sensitization [33,34,35] that enhances pain perception. Increasing evidence suggests that chronic peripheral nociceptive stimuli play a major role in triggering peripheral sensitization first, and then also central sensitization [34], finally resulting in the onset of neural damage and inherent neuropathic pain [36,37]. Therefore, blocking sensitization pathways may be crucial in pain control in KOA management [11]; locally administered BTX has been reported to attenuate primary sensory nerve transduction and transmission, reducing the release of neurogenic inflammatory mediators [38,39] and disrupting the pain-associated receptors [25].

Given that OA is a heterogeneous disease encompassing various distinct phenotypes (i.e., subgroups of patients sharing different and specific pathological mechanisms and related structural and functional manifestations) [40,41], BTX showed a marked and peculiar effect in treating nociceptive pain compared with others (neuropathic or mixed pain). Based on these findings, patients’ phenotyping has been suggested for tailoring treatments based on the specific mechanism of pain pathogenesis.

In conclusion, the RCTs currently available compared BTX with many different approaches for the treatment of knee OA with overall conflicting findings and no evidence of clear BTX superiority to any of the comparators. Nevertheless, the present authors believe that the use of BTX for knee OA should not be discouraged given the favorable safety profile and the encouraging results, but at present, it cannot be preferred or recommended over other approaches. The lack of well-designed RCTs is the main culprit, and this further testifies the fact that low-quality evidence is detrimental both for the scientific community and for patients. Regardless, the data retrieved suggests that BTX is particularly efficient in treating patients who already developed pain sensitization.

#### Limitations

The present manuscript presents some flaws. First of all, a meta-analysis of the data was not performed; the only possible attempt in this regard could have been to compare BTX to hyaluronic acid or the placebo. It could not be possible to perform one because of multiple reasons, particularly the low number of trials, the different clinical score adopted in the studies, non-comparable periods of follow-up in different populations, and the poor homogeneity of data, so that the results would have been unreliable. Furthermore, despite being a systematic review of RCTs, the modest quality of the trials prevents the authors from defining clear indications on the comparative efficacy of BTX versus other conservative approaches.

## 4. Conclusions

The analysis of the available RCTs on BTX intra-articular injection for the treatment of knee OA revealed an overall modest methodological quality. However, based on the data retrieved, botulinum toxin has proven to provide good short-term outcome, especially in patients with pain sensitization. The mechanisms of action include the modulation of neurotransmitter release, peripheral nociceptive transduction, and the control of chronic pain arisen from central sensitization. Further insights are needed to properly profile patients who could benefit more from this peculiar injective approach.

## Figures and Tables

**Figure 1 ijms-24-01486-f001:**
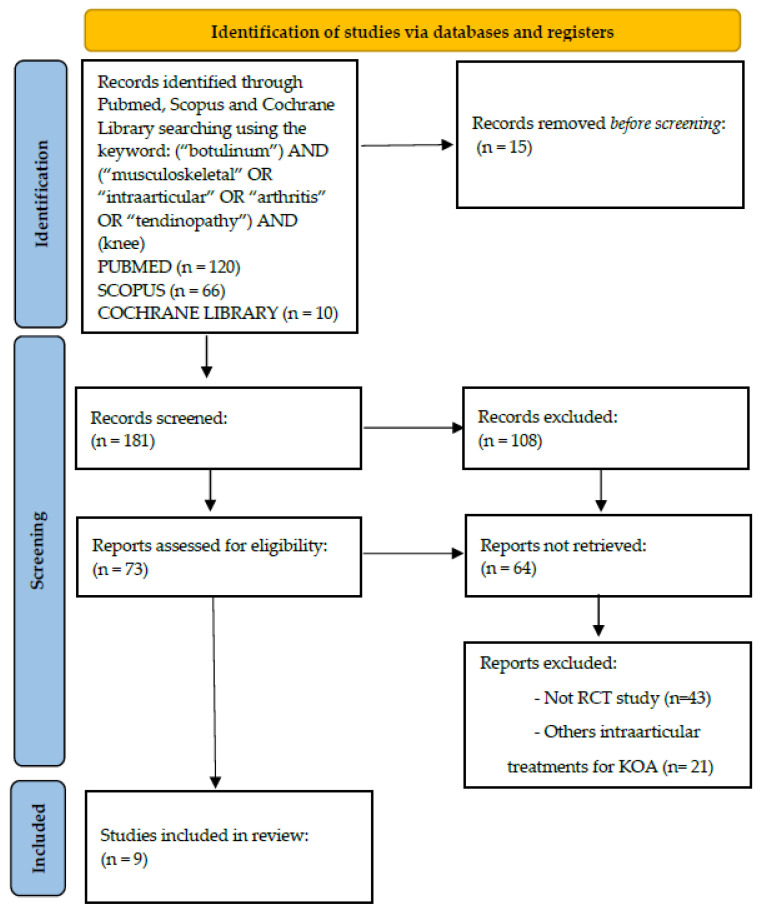
Preferred Reporting Items for Systematic Reviews and Meta-analyses flowchart resuming the paper’s selection process.

**Table 2 ijms-24-01486-t002:** Quality Assessment of the Included Studies by using the Cochrane Risk of Bias tool for Randomized Controlled Trials and the AHRQ (Agency for Healthcare Research and Quality) Standards.

Publication	Random Sequence Generation	Allocation Concealment	Selective Reporting	Other Bias	Blinding of Participants and Personnel	Blinding of Outcome Assessment	Incomplete Outcome Data	AHRQ Standard
Rezasoltani et al., 2021 [28]	Low	Low	Low	Unclear	High	Low	Low	Fair
Rezasoltani et al., 2020 [5]	High	Unclear	Unclear	Unclear	High	High	Low	Poor
Mendes et al., 2019 [20]	Low	Low	Low	Low	Low	Low	Low	Good
McAlindon et al., 2018 [11]	High	High	Unclear	High	Low	Low	Unclear	Poor
Bao et al., 2018 [30]	Low	Unclear	Unclear	Unclear	High	Unclear	Unclear	Poor
Hsieh et al., 2016 [29]	Low	Low	Unclear	High	High	Unclear	Low	Fair
Arendt-Nielsen et al., 2016 [25]	Low	Low	Unclear	Unclear	Low	Low	Unclear	Fair
Boon et al., 2010 [27]	Low	Low	Low	Low	Low	Low	Low	Good
Mahowald et al., 2009 [26]	Unclear	Unclear	Unclear	High	Unclear	Unclear	Unclear	Poor

“Good quality”: All criteria met (i.e., low for each domain); “Fair quality”: One criterion not met (i.e., high risk of bias for one domain) or 2 criteria, and the assessment that this was unlikely to have biased the outcome, and there is no known important limitation that could invalidate the results; “Poor quality”: One criterion not met (i.e., high risk of bias for one domain) or 2 criteria unclear, and the assessment that this was likely to have biased the outcome, and there are important limitations that could invalidate the results; Poor quality: two or more criteria listed as high or unclear risk of bias.

## Data Availability

All the data collected for the purposes of the review have been included in the final manuscript.

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
