# Peer review of "Intra-Articular Injection of Botulinum Toxin for the Treatment of Knee Osteoarthritis: A Systematic Review of Randomized Controlled Trials"

_ijms, 2023, doi:10.3390/ijms24021486_

Round 1
Reviewer 1 Report
This is an interesting systematic review aiming to analyze the available evidences on intra-articular botulinum toxin (BTX) injection in the treatment of knee osteoarthritis, and to compare it to other conservative treatment options. The authors selected papers from the main databases conducting the search for English articles published up to the end of July 2022 according to PRISMA principles, using adequate key words following inclusion criteria for selection: 1) randomized controlled trials (RCTs); 2) written in English language; 3) published on indexed journals from 2001 to 2022; 4) dealing with the use of intra-articular botulinum neurotoxin for the treatment of KOA. The exclusion criteria were: 1) non-randomized trials; 2) reviews; 3) papers written in other languages than English; 4) data not dealing with the treatment of knee osteoarthritis. Nine studies, involving a total of 811 patients with KOA were included. The mean age was 65 years. In most papers, the diagnosis of KOA was assessed using the American College of Rheumatology criteria. Baseline and follow-up assessments were based on clinical and radiological scores. The most used clinical scores were: the Visual Analogue Scale (VAS), the Knee Injury and Osteoarthritis Outcome Score (KOOS), and the Western Ontario and McMaster Universities Osteoarthritis Index (WOMAC). Two trials reported also X-ray and MRI outcomes; one trial included the assessment of synovial hypertrophy, measured by ultrasonography. Risk of bias was analyzed adequately and those studies were detailed. The main findings of the present systematic review were: 1) overall "fair" quality of studies comparing the use of botulinum toxin with other treatments, 2) the lack of univocal results for the intra-articular use of BTX in the treatment of KOA. Due to a small number and heterogeneity of studies the authors did not performed a meta-analysis. Despite this importante limitation, I agree with the authors that the use of BTX should not be discouraged once, based on the data retrieved, botulinum toxin has proven to provide good short-term outcome in patients with pain sensitization. So I suggest to accept this paper for publication, with just minor revision in the english language
Author Response
Thank for your comment. As you suggested, we submitted our work to a native English speaker in order to improve the quality of the language.
Reviewer 2 Report
The work of Sconza et al. is a review that studies the clinical impact of knee osteoarthritis treatment based in the intra-articular injection of botulinum toxin. For achieving this, the work employs data bases such as PubMed, Scopus and Cochrane. The total amount of studies under the inclusion criteria was 9. The authors declare a modest methodology quality in the analysis of available randomized controlled trials on botulim toxin intra-articular injection for the treatment of knee osteoarthritis. However, the authors conclude that botulim toxin has proven to provide good short term out-come, specially in patients with pain sensization.
Questions:
-In order to gain a larger population size that allow for greater statistical power, haven't the authors explored the inclusion in the study of other articles related to the topic, obtained from other databases or published in other languages?
-To what extent does a descriptive work like the one presented by the authors have a clinical predictive value in the absence of any meta-analysis?
Author Response
Thank you for your comment.
- As suggested, in order to obtain a larger population, we expanded the search for randomized controlled trials on the topic by including additional databases, and in particular Web of Science, Pedro, and Research Gate. We also made sure to widen the search to include articles in different languages, such as Italian and Russian. Unfortunately, we found no new RCTs, other than those already mentioned in our review. An additional search was performed on nonconventional databases (i.e., the so-called "gray literature") without detecting any studies that could be included in our systematic review.
- We agree with you that the lack of a meta-analysis could limit the clinical predictive value of the study. In the paragraph “limitation” we highlight the flaws of our systematic review, and the lack of a meta-analysis was one of them. It could not be possible to perform one because of multiple reasons, particularly the low number of trials, the different clinical score adopted in the studies, non-comparable period of follow-up in different populations, so that the results would have been unreliable. Anyway, we now better underlined this limitation in our text (lines 326-329).
In spite of that, we think that our systematic review still shows interesting results, underlining a potentially clinically relevant application for botulin toxin, i.e patients with pain sensitization.
Round 2
Reviewer 2 Report
With all due respect to the authors, I achnowledge the effort to make the study more extensive; however, I am concerned that the lack of a meta-analysis makes the study look like a mere bibliographical work that detracts from its reception by the journal's audience.